# Initial Experience with Left Bundle Branch Area Pacing with Conventional Stylet-Driven Extendable Screw-In Leads and New Pre-Shaped Delivery Sheaths

**DOI:** 10.3390/jcm11092483

**Published:** 2022-04-28

**Authors:** Kyeongmin Byeon, Hye Ree Kim, Seung-Jung Park, Young Jun Park, Ji-Hoon Choi, Ju Youn Kim, Kyoung-Min Park, Young Keun On, June Soo Kim

**Affiliations:** 1Division of Cardiology, Department of Internal Medicine, Chung-Ang University Gwangmyeong Hospital, Chung-Ang University College of Medicine, Gwangmyeong 14353, Korea; kmbyeon79@gmail.com; 2Division of Cardiology, Department of Internal Medicine, Gyeongsang National University Hospital, Gyeongsang National University School of Medicine, Jinju 52727, Korea; hrmanse@naver.com; 3Division of Cardiology, Department of Internal Medicine, Heart Vascular and Stroke Institute, Samsung Medical Center, Sungkyunkwan University School of Medicine, Seoul 06351, Korea; isaturni1985@gmail.com (J.-H.C.); kzzoo921@gmail.com (J.Y.K.); bkm1101@hanmail.net (K.-M.P.); yk.on@samsung.com (Y.K.O.); js58.kim@samsung.com (J.S.K.); 4Division of Cardiology, Department of Internal Medicine, Wonju Severance Christian Hospital, Yonsei University Wonju College of Medicine, Wonju 26426, Korea; pyj@yonsei.ac.kr

**Keywords:** left bundle branch area pacing, pacemaker, stylet-driven extendable screw lead, efficacy, safety

## Abstract

Until recently, left bundle branch area pacing (LBBAp) has mostly been performed using lumen-less fixed screw leads. There are limited data on LBBAp with conventional style-driven extendable screw-in (SDES) leads, particularly data performed by operators with no previous experience with LBBAp procedures. In total, 42 consecutive patients undergoing LBBAp using SDES leads and newly designed delivery sheaths (LBBAp group) were compared with those treated with conventional right ventricular pacing (RVp) for atrioventricular block (RVp group, *n* = 84) using propensity score matching (1:2 ratio). The LBBAp was successful in 83% (35/42) of patients, with satisfactory pacing thresholds (0.8 ± 0.2 V at 0.4 ms). In the LBBAp group, the mean paced-QRS duration obtained during RV apical pacing (173 ± 18 ms) was significantly reduced by LBBAp (116 ± 14 ms, *p* < 0.001). Compared with the RVp group, the LBBAp group showed more physiological pacing, suggested by a much narrower paced-QRS duration (116 ± 14 vs. 151 ± 21 ms, *p* < 0.001). The pacing threshold was comparable in both groups. The LBBAp group revealed stable pacing thresholds for 6.8 ± 4.8 months post-implant and no serious complications including lead dislodgement or septal perforation. The novel approach of LBBAp using SDES leads and the new dedicated pre-shaped delivery sheaths was effectively and safely performed, even by inexperienced operators with LBBAp procedures.

## 1. Introduction

Cardiac conduction system pacing such as His-bundle pacing (HBp) and left bundle branch area pacing (LBBAp) is considered a more physiological treatment for patients with symptomatic bradycardia compared with conventional right ventricular pacing (RVp), which is frequently associated with ventricular dyssynchrony, pacing-induced cardiomyopathy, and higher mortality [1,2]. Prior to the introduction of LBBAp, HBp has fervently been investigated first as one of the viable alternatives to RVp [3]. However, widespread use of HBp has been precluded by several limitations, including a relatively low success rate, a delayed rise in capture thresholds leading to a higher revision rate [4], undersensing of ventricular signals, and oversensing of atrial or His signals [5].

Since Huang et al. [6] addressed the feasibility of LBBAp to implant the pacing lead deep into the basal septum of RV in a patient with heart failure and LBB block (LBBB), LBBAp has become an alternative technique for physiological pacing with a higher success rate and more stable lead parameters than those of HBp [7,8,9,10,11].

Most LBBAp procedures have been performed using lumen-less fixed screw (LLFS) pacing leads (SelectSecure 3830, Medtronic Inc., Minneapolis, MN, USA) delivered through pre-shaped or steerable guiding sheaths (C315HIS or C304, Medtronic Inc., Minneapolis, MN, USA [10,12]. Recently, novel techniques for HBp or LBBAp that utilize standard stylet-driven extendable screw-in (SDES) pacing leads and new pre-shaped delivery sheaths have been introduced. However, there are limited data on LBBAp using the new approach, particularly data performed by operators with no previous experience with LBBAp procedures [13,14,15]. In our institute, only the Biotronik LBBAp tools became recently available but not the Medtronic ones, providing a unique situation where the efficacy and safety of the new approach can be evaluated when performed by operators who have not performed any LBBAp procedures. Therefore, in this paper, we describe our initial experience with LBBAp using SDES leads and new pre-shaped delivery sheaths, primarily focusing on short-term efficacy and safety. We also compared the degrees of ventricular dyssynchrony suggested by paced QRS duration (p-QRSD) measured during RVp and LBBAp in the same patients. Additionally, p-QRSD, pacing threshold, and impedance were compared between patients with LBBAp and a matched control group of patients with conventional RVp.

## 2. Materials and Methods

### 2.1. Patient Population

The pre-existing LLFS lead-based LBBAp system has not yet been introduced in our institute. By contrast, the LBBAp system using SDES leads became available at our institute in December 2020. All adult patients (aged ≥ 18 years) undergoing LBBAp procedures were prospectively enrolled (LBBAp group) for the present study. The LBBAp group was matched to a control group with conventional RVp (RVp group) using a propensity score (PS) at a ratio of 1:2. Patients in the RVp group were selected among those who underwent permanent pacemaker (PPM) implantation for symptomatic bradyarrhythmia from January 2019 to February 2022. To compare the immediate post-implant p-QRSD, only patients with a de novo PPM implant for atrioventricular block (AVB) were included in the control RVp group. We excluded patients with sinus node dysfunction (SND) without AVB or replacement procedures (Figure 1). Leadless pacemaker or biventricular pacemaker cases were also excluded from the present study.

### 2.2. LBBAp Procedures

The thickness of the basal interventricular septum was assessed by echocardiography prior to procedures. LBBAp was performed using SDES pacing leads (Solia S pacing lead, Biotronik SE & Co. KG, Berlin, Germany; 53 cm atrial lead and 60 cm ventricular lead) and specifically designed, fixed-curve delivery sheaths (Selectra 3D, Biotronik SE & Co. KG, Berlin, Germany). The Selectra 3D sheaths with three primary curves (small, middle, or large; 40, 55, or 65 mm) were available. The 12-lead electrocardiograms and the intracardiac electrograms from the pacing lead in a unipolar configuration were continuously recorded. The procedure was performed in a similar way as previously described [12,14,15,16,17,18]. In summary, a temporary pacing wire was deployed into the RV apex for backup pacing in patients with AVB. For the first 10 LBBAp procedures, the His signal was mapped using a quadripolar or His-RV catheter inserted via a femoral vein. However, during the remaining LBBAp procedures (*n* = 32), targeting of LBBA was performed using the “9-partition method”, without His signal mapping [17]. If the sheath was positioned optimally into the RV septum, the 5.6-Fr SDES pacing lead was inserted into the sheath, and the 1.8 mm extendable screw was exposed as previously described [14,15,18]. Unipolar pacing was performed to find an ideal lead position showing a paced QRS complex with a wide “W” shape with a notch in the nadir in lead V1 (red circle in Figure 2A). After finding an ideal lead position, the main operator rotated the lead body rapidly 5 or 6 turns clockwise into the RV septum. Then, unipolar pacing was repeated following each 3 to 4 rotations of the leads to examine the changes in paced QRS morphology and pacing parameters. If a small “r” wave began to appear in V1 lead, one or two rotations were carefully made at a time to avoid septal perforation, until qR or rsR’ pattern appeared in V1 (Figure 2A). Upon rotating the SDES lead, if the helix was rewound back in the lead cage, we turned the outer pin of the lead several times inside the sheath until the screw was fully exposed again. After capping the outer pin with a funnel-shaped stylet guide to prevent the rewinding phenomenon, we resumed the screwing attempts. However, if the rewinding of the helix occurred repeatedly, a new septal region was tried. The p-QRSD, left ventricular activation time (LVAT, interval from pacing artifact to peak in V5 or V6), and pacing impedance were recorded primarily using intermittent unipolar pacing (e.g., before and after each step of screwing). Recently, we began to use continuous unipolar pacing over the stylet as well, which allows uninterrupted real-time monitoring of changes in impedance and paced-QRS morphology during screwing, as previously reported [13]. Success criteria of LBBAp were the presence of RBBB morphology in V1 and at least one of the following: (1) p-QRSD < 130 ms, (2) LVAT < 80 ms, or (3) the presence of LBB potential. In addition, LBBAp was subdivided into (1) selective LBBAp when the isoelectric interval between pacing artifact and the onset of deflection of QRS complex was observed by lower pacing outputs (i.e., <1.5 V at 0.4 ms), and LVAT showed minimal change (<10 ms) with higher (i.e., >5 V at 0.4 ms) or lower pacing outputs; (2) nonselective LBBAp if there was no isoelectric interval and LVAT decreased by >10 ms at higher pacing outputs, compared with the values during lower pacing outputs; or (3) left ventricular septal pacing (LVSp) when no isoelectric interval was present, and LVAT showed minimal change during lower or higher pacing outputs [16,19,20]. Contrast (1–2 mL) was injected in the left anterior oblique view to confirm the lead depth in the interventricular septum (Figure 2B). Echocardiography was performed to assess the location of the lead tip and the presence of pericardial effusion (Figure 2C).

### 2.3. Measurement of Paced-QRS Duration and Pacing-Related Variables

After the LBBAp procedures, p-QRSD was measured at a sweep speed of 100 mm/s by LBBAp and by RV apical pacing (RVAp) with a temporary pacing wire (Figure 2E). In patients without a temporary pacing wire, p-QRSD was measured by pacing Solia leads, which were intentionally deployed at RV apex only transiently (without screwing) before performing the LBBAp procedure. The final pacing threshold and impedance in the unipolar and bipolar configurations were documented before patients were transferred to hospital wards. In the RVp group, data on the immediate post-implant p-QRSDs, bipolar pacing thresholds, and impedances were obtained by reviewing electronic medical record systems.

### 2.4. Statistical Analysis

Categorical variables are presented as counts with percentages and were compared using Fisher’s exact test. Continuous variables are presented as means with standard deviations. Change in p-QRSD (during RVAp and LBBAp) or pacing threshold (at baseline and during follow-up) in the same patients was tested using the Wilcoxon signed-rank test (only in the LBBAp group). The PS was estimated using a regression model with the following baseline variables: age, sex, height^2^, body surface area, hypertension, diabetes mellitus, heart failure, myocardial infarction, previous heart surgery, atrial fibrillation/flutter, chronic renal failure, LV ejection fraction (LVEF), and LV end-diastolic dimension. The LBBAp and RVp groups were matched in a 1:2 ratio using a nearest-neighbor model based on PS. The Mann–Whitney test was used for comparison of p-QRSD, pacing threshold, or pacing impedance between the LBBAp and RVp groups. Statistical analyses were performed using the statistical package SPSS 25.0 software (IBM Corporation, Armonk, NY, USA).

## 3. Results

### 3.1. Patient Characteristics

In the LBBAp group, LBBAp was attempted as a first-choice pacing strategy in 42 patients (Table 1). The indication for LBBAp was AVB in 38 (90.5%), SND with LBBB in 1 (2.4%), and SND alone in 3 patients (7.1%). The mean age was 71 ± 16 years, and 22 patients were male (52%). Six patients had a history of either cardiac surgery or intervention. Three of them had developed AVB after transcatheter aortic valve replacement. LVEF was ≥50% in all but three patients (35, 45, and 48%). LBBB and RBBB patterns of QRS morphology were found in 8 and 11 patients, respectively.

We selected the RVp (control) group out of 1046 patients who received PPM implantation between January 2019 and February 2022 (Figure 1). First, patients with generator replacement (*n* = 224), SND without AVB (*n* = 324), a leadless pacemaker (*n* = 28) or a biventricular pacemaker (*n* = 184) were excluded. Of the remaining 244 patients with de novo PPM implantation for AVB, 84 were selected by PS matching and analyzed as the RVp group. Depending on the location of the RV pacing lead (RV septum or apex), the RVp group was subdivided into RVSp (*n* = 53, 63%) or RVAp (*n* = 31, 37%) subgroups. After PS matching, no significant differences in baseline characteristics were observed between the two groups (Table 1).

### 3.2. Acute Procedural and Short-Term Follow-Up Outcomes in the LBBAp Group

Of the 42 patients who underwent LBBAp procedures, 35 (83%) met the criteria of success with satisfactory pacing thresholds (0.8 ± 0.2 V at 0.4 ms pulse width) and impedances (645 ± 103 Ω). The mean p-QRSD by using LBBAp was 116 ± 14 ms, which was significantly reduced, compared with that by using RVAp in the same patients (173 ± 18 ms, *p* < 0.001) (Figure 3A). The mean LVAT was 83 ± 17 ms, and LBB potential was observed in four patients. Of successful LBBAp cases, selective LBBAp, nonselective LBBAp, and LVSp were observed in 12 (34%), 18 (51%), and 5 (14%) patients, respectively. In the seven patients who failed to satisfy the success criteria, the p-QRS complexes showed a QS pattern in V1, with longer p-QRSDs (136 ± 8 ms) and LVATs (101 ± 11 ms) than those values in successful cases. However, even in the seven unsuccessful implant cases, p-QRSDs by using LBBAp were much narrower, compared with those by using RVAp (187 ± 11 ms, *p* < 0.001).

The mean procedure and fluoroscopic times of LBBAp were 88 ± 40 and 28 ± 16 min, respectively. The number of screwing attempts was 2.1 ± 0.9 times per patient, with more attempts in the unsuccessful cases than in the successful cases (2.3 ± 1.2 vs. 1.9 ± 0.9 times per patient, *p* = 0.299). When patients were divided into the first- and second-half groups, the procedure and fluoroscopic times were significantly reduced in the second-half group, compared with the first-half group (Figure 4); the procedure times ranged from 109 ± 40 to 68 ± 27 min, *p* = 0.001, and the fluoroscopic times ranged from 34 ± 22 to 22 ± 10 min, *p* = 0.027, respectively.

For all patients in the LBBAp group, only the left-sided approach was used, and there was no right-sided implant. The primary curves of the delivery sheaths used for the final successful screwing were small-sized (40 mm) in 2 patients, middle-sized (55 mm) in 24 patients, and large-sized (65 mm) in 16 patients. For each patient, 1.5 ± 0.6 delivery sheaths were used.

Two leads were damaged during the procedures. In one case, the extension–retraction function was impaired, suggesting inner coil breakage. In the other case, the screw was overstretched during unscrewing for repositioning. In contrast, in a patient who required four cycles of repositioning (screwing/unscrewing), the screw remained intact until the final successful implant.

There was no septal perforation, lead dislodgement, or new-onset pericardial effusion before discharge and during a short-term follow-up of 6.8 ± 4.8 months, except for one case of dislodgement that occurred during sheath slitting. Follow-up pacing threshold was measured in 33 of 42 patients in the LBBAp group and it remained low and stable; baseline vs. follow-up, 0.8 ± 0.2 vs. 0.9 ± 0.2 V at 0.4 ms, *p* = 0.132.

### 3.3. Comparison of Acute Outcomes between the LBBAp and RVp Groups

When compared with the RVp group (*n* = 84), the LBBAp group (*n* = 42) showed more physiological ventricular pacing, suggested by a much narrower-paced QRS duration (116 ± 14 vs. 151 ± 21 ms, *p* < 0.001). Even if the RVp group was divided into RVAp (*n* = 31) and RVSp (*n* = 53) subgroups, the p-QRSD was significantly shorter in the LBBAp than in either of the RVp subgroups (Figure 3B). In addition, baseline LBBB was corrected in seven of eight patients in the LBBAp group. Their QRSD significantly decreased from 176 ± 18 to 113 ± 15 ms (*p* = 0.001), suggesting that more physiological ventricular contraction was achieved by using LBBAp.

In contrast, the LBBAp group showed comparable performance to the RVp group in terms of immediate post-implant pacing threshold (0.8 ± 0.2 vs. 0.7 ± 0.2 V at 0.4 ms, *p*= 0.502, Figure 3C). Pacing impedance was lower in the LBBAp than in the RVp groups (645 ± 89 vs. 729 ± 165 Ω, *p* < 0.001, Figure 3D); however, the values were within the acceptable range in both groups.

## 4. Discussion

### 4.1. Main Findings

The main findings of the present study were as follows: (1) the success rate of LBBAp using SDES leads and new pre-shaped sheaths was high (83%, 35/42), with satisfactory pacing thresholds (0.8 ± 0.2 V at 0.4 ms) even by operators who had not performed any LBBAp procedure using LLFS or SDES leads, and the procedure/fluoroscopic times were significantly reduced with repeated procedures; (2) short-term safety of LBBAp with this new approach was demonstrated without significant complications such as septal perforation, lead dislodgement, pericardial effusion, or pneumothorax; and (3) more physiological ventricular pacing suggested by narrower p-QRSDs was achieved by LBBAp, compared with RVp, without compromising pacing threshold or impedance.

### 4.2. Efficacy and Safety of LBBAp Using SDES Leads and New Fixed Curve Delivery Sheath

Previously, conduction system pacing such as HBp or LBBAp was performed using LLFS leads and Medtronic pre-shaped (C315HIS) or steerable guiding sheaths (C304) [7,8,9,10,11,12,21]. Recently, a new approach using SDES leads and Selectra 3D sheaths was introduced for the same purpose. However, limited data are available on HBp/LBBAp using this new method. Zingarini et al. showed that HBp performed using this new system was feasible and safe in 17 consecutive patients with SSS (59%) or AVB (41%) with acceptable pacing-related parameters [22]. Regarding the LBBAp, only several studies are available to date [13,14,15]. In a nonrandomized study by De Pooter et al., the success rate of LBBAp was 87% (20/23) with SDES leads and 89% (24/27) with LLFS leads (*p* = 0.834), without lead revision. Additionally, screwing attempts, screw implant depth, procedural/fluoroscopy times, and bipolar pacing thresholds (0.6 ± 0.1 vs. 0.5 ± 0.3 V at 1.0 ms pulse width, *p* = 0.466) were comparable between the two lead systems. In line with the previous studies, our results indicate that LBBAp using SDES leads and new fixed sheaths was effective and safe with implant success rate, p-QRSD, and pacing thresholds or impedances comparable to those of previous reports.

Our bipolar pacing threshold (0.8 ± 0.2 V) seemed to be slightly higher than the previous data (0.6 ± 0.1 V) [14]. However, our data were obtained at a shorter pulse width (0.4 ms) than that of the previous study (1.0 ms). Our relatively high success rate despite lack of previous experience with the LBBAp or HBp procedures might be associated with the much wider structure of the LBB, compared with the thin HB [23]. Additionally, the structural similarity between the new Selectra 3D sheaths and the former LV lead delivery sheaths might help operators become accustomed quickly to the LBBAp procedures using the new tools. Our fluoroscopic and procedural times were longer than those values from the currently published literature. This gap is probably related to our insufficient experience with this new procedure. However, those times significantly decreased with repeated procedures (Figure 4). Particularly, procedural and fluoroscopic times in the last quarter of LBBAp procedures were likely to decrease further (*n* = 11, procedural time 55 ± 22 min, and fluoroscopic time 18 ± 10 min). Most of our patients had impaired LBB conduction (AVB or LBBB), and we did not use the “dual lead technique” developed by Huang et al., which probably led to the lower rate of LBB potential recording (9.5%, 4/42) in our study than that in the previous report [14]. The mean number of sheaths used in this study (1.5 ± 0.6 per patient) may be higher than expected. This may also be due to our previous lack of experience with LBBAp procedures and new tools. In addition, no criteria have been proposed to guide optimal sheath selection. Therefore, it may be worth conducting further studies to find the criteria for selecting the optimal sheath size.

Although the SDES lead (5.6 Fr) has a slightly greater diameter than the LLSF lead (4.1 Fr), screwing the SDES leads into the septal myocardium was effective without causing septal perforation. However, there were two cases where the extendable–retractable screw was broken during lead repositioning. Therefore, careful handling of the SDES leads may be needed during repositioning. In addition, simple recommendations to avoid this risk were previously reported by le Polain de Waroux et al. as follows: do not further rotate the lead if no progression is observed, never remove the stylet insertion tool before the lead reaches its final position, and never try to unscrew the helix before repositioning [15].

### 4.3. Efficacy of Physiological Pacing Using the New Implant Approach

The RVAp-induced QRS widening is one of the well-known risk factors for LV dyssynchrony, pacing-induced cardiomyopathy, and long-term poor prognosis [1,2,24]. Therefore, non-RVAp strategies such as RVSp or RV outflow track pacing were attempted to reduce the p-QRSD and minimize the risk of RVAp-related adverse outcomes [25,26]. Our data showed that LBBAp was superior to RVp, regardless of RVSp or RVAp, in lowering the p-QRSD (Figure 3B). Moreover, even in our unsuccessful LBBAp cases, p-QRSDs were much shorter than those obtained during RVAp (Figure 3A). In addition, baseline LBBB disappeared by LBBAp in seven of our eight patients, which probably exerts beneficial long-term effects on ventricular function.

### 4.4. Limitations

We acknowledge that the present study has several limitations. First, a small number of patients was analyzed for a short duration. Therefore, we were not able to investigate specific factors to help operators select an optimal-sized primary curve of the Selectra 3D sheaths for each patient. However, our data are one of the largest LBBAp cohorts performed using the new approach. In addition, there was no chance to perform the right-sided subclavian implantation; accordingly, we cannot provide data on the right-sided approach. In a previous study, repetitive lead dislodgment was reported while slitting the right-sided delivery sheath. The authors also experienced the rotation of the lead being hindered by the over-torqued right-sided sheath [14]. Therefore, more data may be needed on the success rate of the right-sided approach. In addition, a dedicated right-sided sheath might be needed to facilitate the right-sided implant. Another limitation was that the effect of LBBAp versus RVp on ventricular function, morbidity, and mortality was not evaluated. Therefore, prospective studies on a larger scale are needed to validate the long-term efficacy and safety of LBBAp using this new approach. Finally, LBBAp using the two systems (SEDS versus LLFS leads) was not compared in our data.

## 5. Conclusions

LBBAp using stylet-driven extendable screw-in pacing leads and the new pre-shaped delivery sheaths is feasible and safe without acute, significant complications, even when performed by operators with no previous experience with LBBAp procedures. LBBAp provided more physiological ventricular pacing, compared with conventional RVp, with remarkably narrower p-QRSD without compromising pacing threshold or impedance.

## Figures and Tables

**Figure 1 jcm-11-02483-f001:**
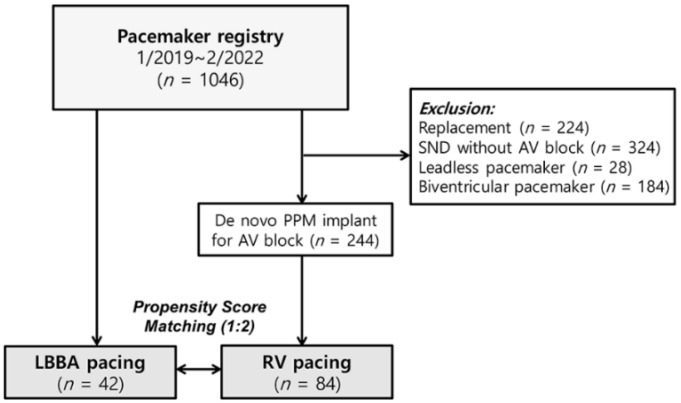
Flowchart of enrollment. AV, atrioventricular. LBBA, left bundle branch area. PPM, permanent pacemaker. RV, right ventricle. SND, sinus node dysfunction.

**Figure 2 jcm-11-02483-f002:**
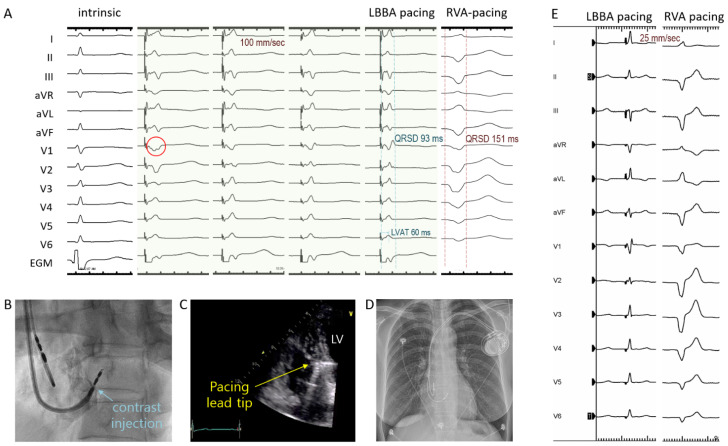
Twelve-lead surface electrocardiogram and intracardiac electrogram (EGM) during intrinsic rhythm, unipolar LBBA pacing, and RVA pacing (**A**). Ideal location of LBBA shows a wide “W” shape QRS complex in V1 lead (red circle). Contrast injection (**B**) and echocardiography (**C**) showing the pacing lead tip implanted into the interventricular septum. After implantation of the LBBA pacing system (**D**), much narrower QRS duration was achieved by LBBA pacing (**Left**), compared with RVA pacing (**Right**) in the same patient (**E**). LBBA, left bundle branch area; LVAT, left ventricular activation time; QRSD, QRS duration; RVA, right ventricular apex; LV, left ventricle.

**Figure 3 jcm-11-02483-f003:**
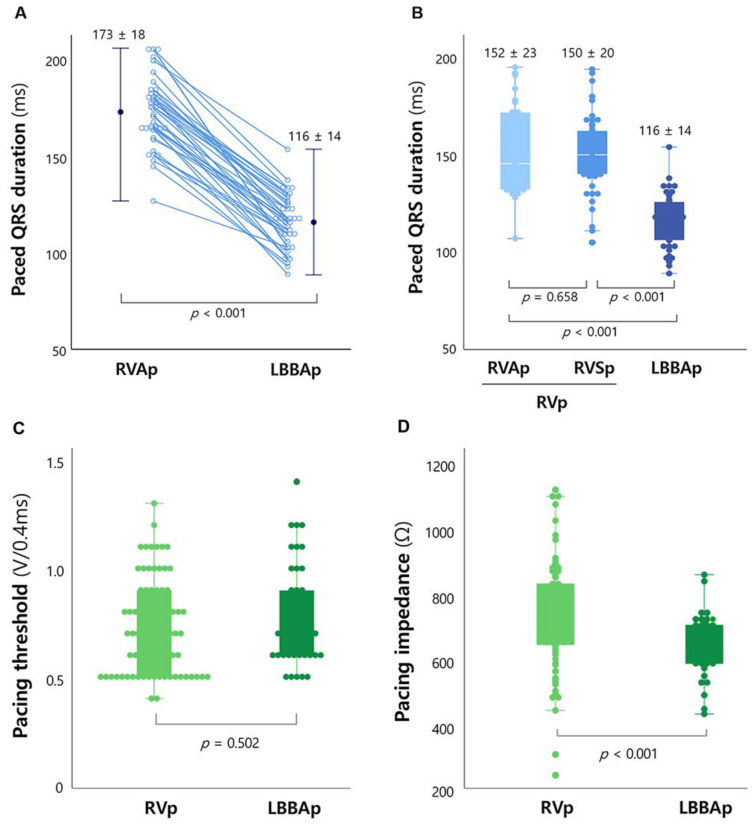
Paced QRS durations according to pacing location (RVAp vs. LBBAp) in the same patients (**A**). Comparison of paced QRS duration (**B**), pacing threshold (**C**), and pacing impedance (**D**) in the LBBAp and RVp groups. The RVp group was divided into the RVAp and RVSp subgroups (**B**). LBBAp, left bundle branch area pacing; p-QRSD, paced QRS duration; RVp, right ventricular pacing; RVAp, RV apical pacing; RVSp, RV septal pacing.

**Figure 4 jcm-11-02483-f004:**
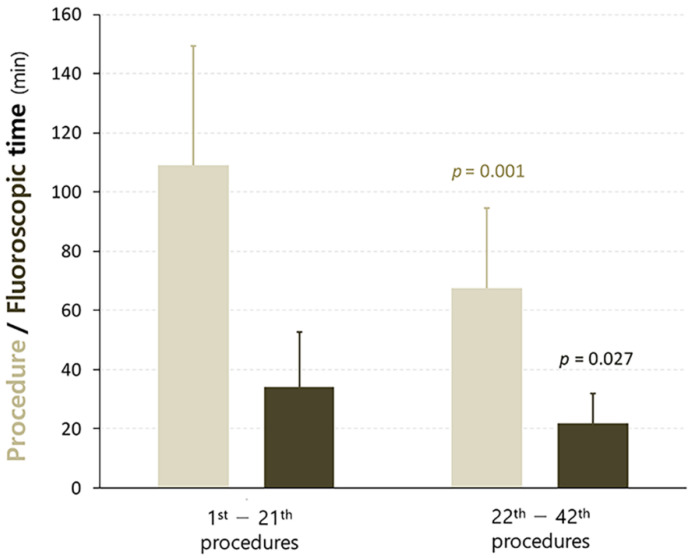
Change in procedure/fluoroscopic times according to LBBA pacing procedural experience. LBBA, left bundle branch area.

**Table 1 jcm-11-02483-t001:** Baseline patient characteristics.

	LBBA Pacing (*n* = 42)	Non-LBBA Pacing (*n* = 84)	*p* Value
*Demographics*			
Age (years)	71 ± 16	69 ± 15	0.232
Male	22 (52)	45 (54)	0.900
Height	2.54 ± 0.29	2.58 ± 0.33	0.475
BSA (m^2^)	1.62 ± 0.17	1.65 ± 0.19	0.504
*Medical history*			
Diabetes mellitus	15 (36)	30 (36)	1.000
Hypertension	20 (48)	43 (51)	0.705
Heart failure	10 (24)	9 (11)	0.053
Myocardial infarction	1 (2)	0 (0)	0.333
Previous surgery/intervention	6 (14)	11 (13)	0.854
Chronic kidney disease	5 (12)	5 (6)	0.244
*Echo- and Electrocardiographic variables*			
LVEF (%)	62 ± 9	63 ± 7	0.436
LVEDD (mm)	52 ± 6	51 ± 5	0.932
AF/AFL	9 (21)	13 (16)	0.407
*Pacemaker-related variables*			
Indication of pacing	AVB, 38 (90)	AVB, 84 (100)	
	SSS, 4 (10)	SSS, 0 (0)	
Pacing location	LBBA, 42 (100)	RV apex, 31 (37)	
		RV septum *, 53 (63)	

Values are expressed as *n* (%), or mean ± SD. * indicates non-LBBA conventional RV septum. AF = atrial fibrillation, AFL = atrial flutter, AVB = atrioventricular block, BSA = body surface area (Du Bois formula), LBBA = left bundle branch area, LVEDD = left ventricular end diastolic dimension, LVEF = left ventricular ejection fraction, RV = right ventricle, SSS = sick sinus syndrome.

## Data Availability

All data generated or analyzed during this study are included in this article. Further inquiries can be directed to the corresponding author.

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
