# Peer review of "Initial Experience with Left Bundle Branch Area Pacing with Conventional Stylet-Driven Extendable Screw-In Leads and New Pre-Shaped Delivery Sheaths"

_jcm, 2022, doi:10.3390/jcm11092483_

Round 1
Reviewer 1 Report
Byeon et al present their experience on LBBAP using stylet-driven pacing leads. Prospective, single center study, n=42, good implant success, adequate pacing thresholds and no severe lead complications. Comparison with RV-pacing using propensity score matching.
My comments:
- upon rotating SDL, the helix might rewind back in the lead cage, how did the authors deal with this?
- the mean number of sheaths used in this study was 1.5/patient, this seems high. Please explain.
- the used criteria to define LBBAP, seem not to allow any differentiation between myocardial capture (Left ventricular septal pacing) and non-selective LBBP. Please add data on number of patients with proven conduction system capture and pure myocardial capture.
Author Response
Response letter for the manuscript “Initial Experience with Left Bundle Branch Area Pacing with Conventional Stylet-Driven Extendable Screw-in Leads and New Pre-shaped Delivery Sheaths”
April/21, 2022
Dear Emmanuel Andrès and Reviewer #1
Thank you very much for your review and the helpful comments of the reviewer about our manuscript. We appreciate the opportunity to resubmit our revised manuscript.
In accordance with the constructive comments suggested by the reviewers, we made every effort to address the many of the reviewers’ comments, revising our manuscript.
The revised parts of the manuscript are highlighted in blue font for your convenience. We believe that these revisions strengthen our manuscript and hope that this better meets the requirements of your journal. We really hope that the revised manuscript is now suitable for the publication in Journal of Clinical Medicine.
Thank you again for your constructive and detailed review on our manuscript.
Yours sincerely,
Seung-Jung Park, MD
Division of Cardiology, Department of Medicine, Heart Vascular Stroke Institute
Samsung Medical Center, Sungkyunkwan University School of Medicine
Tel: 82-2-3410-7145, Fax: 82-2-3410-3849
Email: orthovics@skku.edu, orthovics@gmail.com
Response to Reviewer’s comments
Reviewer #1’s comments:
Byeon et al present their experience on LBBAP using stylet-driven pacing leads. Prospective, single center study, n=42, good implant success, adequate pacing thresholds and no severe lead complications. Comparison with RV-pacing using propensity score matching.
Comment 1
upon rotating SDL, the helix might rewind back in the lead cage, how did the authors deal with this?
<Response>
First of all, we would like to express our deepest gratitude to the reviewer for taking the effort and time for reviewing our manuscript.
As you mentioned, we have experienced several times that the helix of the SDL was rewound back into the lead cage while attempting to rotate it, although the outer pin was capped with a funnel-shaped stylet guide to prevent rewinding of the helix before the screwing attempt. Whenever the rewinding was noticed on fluoroscopy, we turned the outer pin of the lead several times inside the sheath until the 1.8-mm extendable screw was fully exposed again, and recapped the outer pin with the stylet guide and continued the screwing procedure. However, if the rewinding of the helix occurred repeatedly, a new septal region was tried for the screwing procedure. To reflect the importance of your comments, we added this point into the method section of the revised manuscript as follows;
Upon rotating the SDES lead, if the helix was rewound back in the lead cage, we turned the outer pin of the lead several times inside the sheath until the screw was fully exposed again. After capping the outer pin with a funnel-shaped stylet guide to prevent the rewinding phenomenon, we resumed the screwing attempts. However, if the rewinding of the helix occurred repeatedly, a new septal region was tried.
Thank you so much for your comments.
Comment 2
the mean number of sheaths used in this study was 1.5/patient, this seems high. Please explain.
<Response>
We understand your concerns about using a larger number of sheaths than expected for our patients. This was probably related with our previous lack of experience with the LBBA pacing procedures and new tools. Although several factors including height, overall heart size, and left atrial diameter were taken into account while selecting the initial sheath size, it was often needed to change sheath size for successful procedures. It may be worth conducting further studies to find criteria for selecting the optimal initial sheath size. Thank you so much for your insightful comments.
Considering the importance of your comments, we added this point into the discussion section as follows;
The mean number of sheaths used in this study (1.5 ± 0.6 per patient) may be higher than expected. This may also be due to our previous lack of experience with LBBAp procedures and new tools. In addition, no criteria have been proposed to guide optimal sheath selection. Therefore, it may be worth conducting further studies to find criteria for selecting the optimal sheath size.
Comment 3
the used criteria to define LBBAP, seem not to allow any differentiation between myocardial capture (Left ventricular septal pacing) and non-selective LBBP. Please add data on number of patients with proven conduction system capture and pure myocardial capture.
<Response>
These are excellent and very important points that were well taken by all the authors. We totally agree with your opinion and acknowledge that our initial description was not sufficient to differentiate left ventricular septal pacing (LVSp) and non-selective LBBAp. Therefore, we provided further description on the definition of the subtypes of LBBAp in the method section of our revised manuscript as follows;
Success criteria of LBBAp were the presence of RBBB morphology in V1 and at least one of the following: (1) p-QRSD < 130 ms, (2) LVAT < 80 ms, or (3) the presence of LBB potential. In addition, LBBAp was subdivided into (1) selective LBBAp when isoelectric interval between pacing artifact and the onset of deflection of QRS complex was observed by lower pacing outputs (< 1.5 V at 0.4ms) and LVAT showed minimal change (<10 ms) with higher (> 5 V at 0.4 ms) or lower pacing outputs; (2) nonselective LBBAp if there was no isoelectric interval and LVAT decreased by >10 ms at higher pacing outputs compared to the values during lower pacing outputs; or (3) left ventricular septal pacing (LVSp) when no isoelectric interval was present and LVAT showed minimal change during lower or higher pacing outputs.
According to the above-mentioned definition, selective LBBAp, nonselective LBBAp, and LVSp were observed in 12 (34%), 18 (51%), and 5 (14%) patients of our successful LBBAp cases, respectively. We have added the detailed outcomes into the result section as well.
Thank so much for your careful and constructive comments.

Reviewer 2 Report
Dear Editor dear Authors, thank you for the opportunity to review the article „Initial Experience with Left Bundle Branch Area Pacing with Conventional Stylet-Driven Extendable Screw-in Leads and New Pre-shaped Delivery Sheaths“
In general this is a very interesting study with a very important topic, the learning curve, which is a important massage to our colleagues around the world. Congratulations to the authors. I have following questions:
- It is not clear in the methods (in text - but described in Figure 2) whether the presence of a “W” configuration of the QRS complex in lead V1 was your indicator for screwing target selection. Please describe it clearly.
- We also do a lot of LBBAP with stylet driven leads and use continuous unipolar pacing over the stilet during screwing, which seems to be one of the biggest benefits. If I understand you correctly, you are looking for unipolar stimulation after every 3-4 screwings, could you please comment/correct this point?
- Please provide information on the average number of screwing attempts in successful and unsuccessful cases.
- Please provide information on the criteria for choosing a medium or large curve, whether it is based on pre-procedure measurements.
Best Regards
Author Response
Response letter for the manuscript “Initial Experience with Left Bundle Branch Area Pacing with Conventional Stylet-Driven Extendable Screw-in Leads and New Pre-shaped Delivery Sheaths”
April/21, 2022
Dear Emmanuel Andrès and Reviewer #2
Thank you very much for your review and the helpful comments of the reviewer about our manuscript. We appreciate the opportunity to resubmit our revised manuscript.
In accordance with the constructive comments suggested by the reviewers, we made every effort to address the many of the reviewers’ comments, revising our manuscript.
The revised parts of the manuscript are highlighted in blue font for your convenience. We believe that these revisions strengthen our manuscript and hope that this better meets the requirements of your journal. We really hope that the revised manuscript is now suitable for the publication in Journal of Clinical Medicine.
Thank you again for your constructive and detailed review on our manuscript.
Yours sincerely,
Seung-Jung Park, MD
Division of Cardiology, Department of Medicine, Heart Vascular Stroke Institute
Samsung Medical Center, Sungkyunkwan University School of Medicine
Tel: 82-2-3410-7145, Fax: 82-2-3410-3849
Email: orthovics@skku.edu, orthovics@gmail.com
Reviewer #2’s comments:
In general this is a very interesting study with a very important topic, the learning curve, which is a important massage to our colleagues around the world. Congratulations to the authors. I have following questions:
Comment 1
It is not clear in the methods (in text - but described in Figure 2) whether the presence of a “W” configuration of the QRS complex in lead V1 was your indicator for screwing target selection. Please describe it clearly.
<Response>
First of all, we would like to express our deepest gratitude to the reviewer for taking the effort and time for reviewing our manuscript.
As you commented, the presence of a “W” configuration of the QRS complex in lead V1 was the indicator for the selection of target site for screwing the leads. According to your kind suggestion, we have added this point into the method section of our revised manuscript as follows;
Unipolar pacing was performed to find an ideal lead position showing a paced QRS complex with a wide "W" shape with a notch in the nadir in lead V1 (red circle in Figure 2A).
Thank you so much for your kind suggestions.
Comment 2
We also do a lot of LBBAP with stylet driven leads and use continuous unipolar pacing over the stilet during screwing, which seems to be one of the biggest benefits. If I understand you correctly, you are looking for unipolar stimulation after every 3-4 screwings, could you please comment/correct this point?
<Response>
We completely agree with your opinion that continuous unipolar pacing over the stylet during screwing is a very useful method, and can provide operators with uninterrupted real-time monitoring of changes in impedance and paced-QRS morphology during screwing as previously reported (J Cardiovasc Electrophysiol 2022;33:299-307). Initially, we have used conventional interrupted unipolar pacing on the connector-pin. However, recently, we began to use the continuous unipolar pacing over the stylet as well.
Thank you so much for your careful review. To reflect the importance of your comments, we added this point in the method section of our revised manuscript as follows;
The paced QRS duration, left ventricular activation time (LVAT, interval from pacing artifact to peak in V5 or V6) and pacing impedance were recorded primarily using intermittent unipolar pacing (e.g., before and after each step of screwing). Recently, we began to use continuous unipolar pacing over the stylet as well, which allows uninterrupted real-time monitoring of changes in impedance and paced-QRS morphology during screwing as previously reported.
Comment 3
Please provide information on the average number of screwing attempts in successful and unsuccessful cases.
<Response>
The average number of screwing attempts in the unsuccessful group (2.3 ± 1.2) seemed to be greater than the value of the successful group (1.9 ± 0.9, P= 0.299). This information would be informative and interesting for some readers. So, we added this into the result section. Thank you so much for your insightful comments.
Comment 4
Please provide information on the criteria for choosing a medium or large curve, whether it is based on pre-procedure measurements.
<Response>
These are excellent and very important points that were well taken by all the authors. We totally agree with your opinion that some criteria for choosing an optimal sized sheath would be very useful. In fact, we have taken into account several factors including height, overall heart size, and left atrial diameter to select optimal sized sheaths. However, it was often needed to change sheath size for successful procedures. Additionally, no criteria have been proposed to guide optimal sheath selection. Therefore, it may be worth conducting further studies to find criteria for selecting the optimal sheath size. Thank you so much for your insightful comments.
Considering the importance of your comments, we added this point into the discussion section.
